# A machine learning-based approach for constructing remote photoplethysmogram signals from video cameras

Rodrigo Castellano Ontiveros [1,2,3], Mohamed Elgendi [1,3] ✉ & Carlo Menon [1] ✉

## Abstract

**Background** Advancements in health monitoring technologies are increasingly relying on capturing heart signals from video, a method known as remote photoplethysmography (rPPG). This study aims to enhance the accuracy of rPPG signals using a novel computer technique.

**Methods** We developed a machine-learning model to improve the clarity and accuracy of rPPG signals by comparing them with traditional photoplethysmogram (PPG) signals from sensors. The model was evaluated across various datasets and under different conditions, such as rest and movement. Evaluation metrics, including dynamic time warping (to assess timing alignment between rPPG and PPG) and correlation coefficients (to measure the linear association between rPPG and PPG), provided a robust framework for validating the effectiveness of our model in capturing and replicating physiological signals from videos accurately.

**Results** Our method showed significant improvements in the accuracy of heart signals captured from video, as evidenced by dynamic time warping and correlation coefficients. The model performed exceptionally well, demonstrating its effectiveness in achieving accuracy comparable to direct-contact heart signal measurements.

**Conclusions** This study introduces a novel and effective machine-learning approach for improving the detection of heart signals from video. The results demonstrate the flexibility of our method across various scenarios and its potential to enhance the accuracy of health monitoring applications, making it a promising tool for remote healthcare.

## Plain language summary

This research explores a new way to monitor health using video, which is less invasive than traditional methods that require direct skin contact. We developed a computer program that improves the accuracy of heart signals captured from video. This is done by comparing these video-based signals with standard clinical signals from physical sensors on the skin. Our findings show that this new method can match the accuracy of conventional clinical methods, enhancing the reliability of non-contact health monitoring. This advancement could make health monitoring more accessible and comfortable, offering a potential for doctors to track patient health remotely, making everyday medical assessments easier and less intrusive.

Remote photoplethysmography (rPPG) is a non-invasive method for detecting volumetric variations in blood using video cameras[1,2]. It provides a promising alternative to contact photoplethysmography (cPPG) methods[3] and has been used in various applications, including heart rate monitoring, stress detection, sleep analysis[4], and hypertension[5]. However, the accuracy of rPPG signals can be affected by factors such as motion artifacts, changes in illumination, and skin tone variations[6,7]. These factors can introduce noise and distortions into the captured signal, leading to inaccurate measurements of physiological parameters. For instance, in the dataset LGI-PPGI, the videos in which the subjects talk are recorded outdoors, impacting the quality of the rPPG. When comparing factors in rPPG to conventional cPPG, certain distinctions emerge. While both methods are susceptible to motion-related artifacts, rPPG faces additional challenges due to the reliance on non-contact measurements, making it more sensitive to environmental changes and subject movement. To avoid these problems, best practices in PPG signal acquisition and processing are described in ref. 8.

[1]Biomedical and Mobile Health Technology Lab, Department of Health Sciences and Technology, ETH Zurich, Zurich, Switzerland. [2]School of Electrical Engineering and Computer Science, KTH Royal Institute of Technology, Stockholm, Sweden. [3]These authors contributed equally: Rodrigo Castellano Ontiveros, Mohamed Elgendi. ✉e-mail: moe.elgendi@hest.ethz.ch; carlo.menon@hest.ethz.ch

Our goal is to improve the quality of the rPPG signal by constructing it similarly to the contact PPG signal. This will allow for more accurate extraction of the physiological information obtained from rPPG signals, such as pulse rate variability[9]. There have been previous attempts to improve physiological parameters derived from rPPG signals. For example, some studies have evaluated the performance of restored rPPG signals by comparing heart rate (HR) and HR variability (HRV) obtained from rPPG with a reference HR and HRV extracted from cPPG[10–13]. Other authors have focused on improving BP[14], oxygen saturation measurements[15], or fibrillation arrhythmia[16].

Most existing studies have focused on computing specific physiological parameters from remote photoplethysmography (rPPG) signals rather than capturing the raw signal itself[14]. However, if a model is trained to obtain an rPPG signal for a specific parameter such as HR, other information may be lost. For instance, only the systolic peak is relevant for HR calculation, resulting in the loss of information about the diastolic peak. In addition, the morphology of the signal itself can provide valuable information, such as first and second derivatives, that can be useful for detecting cardiovascular diseases (CVDs)[17]. Although some studies have achieved great results calculating HR using rPPG signals, these signals are often not robust and contain a high amount of noise[18]. This is because these authors' models have been trained to improve HR calculation rather than to improve the signal itself. One study attempted to improve PPG signals by comparing them to reference PPG signals[19], but the study used a private dataset without activity distinction and evaluated rPPG and contact PPG (cPPG) using Pearson correlation coefficient and cosine similarity. We improve this by including three public datasets and more metrics useful to compare the rPPG with the cPPG, in both time and frequency domains.

In this study, we present a machine learning-based approach to enhance the construction of remote photoplethysmogram (rPPG) signals from video cameras. Our model utilizes existing rPPG signals from various models such as Chrominance (CHROM)-based rPPG[20], Local Group Invariance (LGI)[21], Independent Component Analysis (ICA)[22], and Plane-Orthogonal-to-Skin (POS)[23] as inputs. The choice of these algorithms was based on the recommendation of a previous study[24]. It enhances these by comparing them with reference PPG signals. This novel method marks a significant advancement in rPPG signal detection by employing machine learning techniques, compared to traditional methods that extract features directly from the video followed by classification[25].

## Methods

In this section, the methodology used in this study is presented, from the data processing techniques to the models used to construct the rPPG. A general visualization of the pipeline is presented in Fig. 1.

## Dataset description and ethical compliance

For this study, three public datasets were utilized:

LGI-PPGI: This dataset is published under the CC-BY-4.0 license. The study was supported by the German Federal Ministry of Education and Research (BMBF) under the grant agreement VIVID 01|S15024 and by CanControls GmbH Aachen[21]. The LGI-PPGI dataset is a collection of videos featuring six participants, the sex of five is male and one is female. The participants were recorded while performing four activities: Rest, Talk, Gym (exercise on a bicycle ergometer), and Rotation (rotation of the head of the subject at different speeds). The videos were captured using a Logitech HD C270 webcam with a frame rate of 25 fps, and cPPG signals were collected using a CMS50E PPG device at a sampling rate of 60 Hz. The videos were shot in varying lighting conditions, with talking scenes recorded outdoors and other activities taking place indoors.

PURE: Access to this dataset is granted upon request. It received support from the Ilmenau University of Technology, the Federal State of Thuringia, and the European Social Fund (OP 2007-2013) under grant agreement N501/2009 for the project SERROGA (project number 2011FGR0107)[26]. The PURE dataset contains videos of 10 participants, of which eight have the sex male and two female, engaged in various activities classified as Steady, Talk, Slow Translation (average speed is 7% of the face height per second), Fast Translation (average speed is 14% of the face height per second), Small Rotation (average head angle of 20°), and Medium Rotation (average head angle of 35°). The videos were captured using a 640 × 480 pixel eco274CVGE camera by SVS-Vistek GmbH, with a 30 fps frame rate and a 4.8 mm lens. The cPPG signals were collected using a CMS50E PPG device at a sampling rate of 60 Hz. The videos were shot in natural daylight, with the camera positioned at an average distance of 1.1 m from the participants' faces.

MR-NIRP indoor: This dataset is openly accessible without any restrictions. It received funding under the NIH grant 5R01DK113269-02[27]. The MR-NIRP indoor video dataset is comprised of videos of eight participants, including six participants with sex male and two female, with different skin tones: 1 Asian, 4 Indian, and 3 Caucasian. The participants were recorded while performing Still and Motion activities, with talking and head movements being part of the latter. The videos were captured using a FLIR Blackfly BFLY-U3-23S6C-C camera with a resolution of 640 × 640 and a frame rate of 30 fps. The cPPG signals were collected using a CMS 50D+ finger pulse oximeter at a sampling rate of 60 Hz.

Each dataset includes video recordings of participants engaged in various activities, alongside a reference cPPG signal recorded using a pulse oximeter. Table 1 provides detailed characteristics of each dataset.

**Fig. 1 | General workflow.** From data processing to comparison of the reference photoplethysmogram (PPG) with the remote photoplethysmogram (rPPG) constructed by the model. CV cross-validation, RGB red, green, and blue channels, ML machine learning. Colors: the green signal refers to the rPPG reconstructed by the model, and the black signal refers to the fingertip PPG.

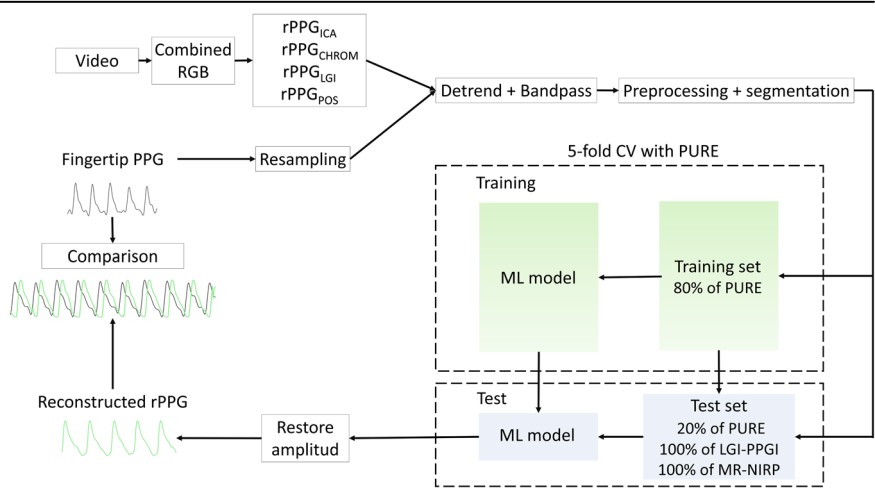

**Table 1 | Description of the LGI-PPGI, PURE, and indoor MR-NIRP datasets**

| Participants | LGI-PPGI 6 | PURE 10 | MR-NIRP indoor 8 |
|---|---|---|---|
| Activities | Resting, Talking, exercising on a bicycle ergometer (Gym), and Rotation | Steady, Talking, Slow Translation, Fast Translation, Small Rotation, and Medium Rotation | Still and Motion |
| Pulse oximeter (sampling rate) | CMS50E (60 Hz) | CMS50E (60 Hz) | CMS 50D+ (60 Hz) |
| Camera (fps) | Logitech HD C270 webcam (25 fps) | Eco274CVGE (30 fps) | FLIR Blackfly BFLY-U3-23S6C-C (30 fps) |

### Ethical considerations for secondary use

The datasets used in our research are not only publicly available but are also extensively utilized within the scientific community for various secondary analyses. All datasets received the requisite ethical approvals and informed consents, in accordance with the regulations of their respective academic institutions. This compliance facilitated the publication of the data in academic papers and its availability online. The responsibility for managing ethical compliance was handled by the original data providers. They ensured that these datasets were made available under terms that permit their use and redistribution with appropriate acknowledgment.

Given the extensive use of these datasets across multiple studies, additional IRB approval for secondary analyses of de-identified and publicly accessible data is typically not required. This practice aligns with the policies at ETH Zurich, which do not mandate further IRB approval for the use of publicly available, anonymized data.

A comprehensive description of each dataset, including its source, funding agency, and licensing terms, has been provided in the manuscript. This ensures full transparency and adherence to both ethical and legal standards.

### rPPG extraction from video

Several steps were necessary to extract the rPPG signal from a single video. First, the regions of interest (RoI) were extracted from the face. We extracted information from the forehead and cheeks using the pyVHR framework[28], which includes the software MediaPipe for the extraction of RoI from a human face[29]. The RoI extracted from every individual were composed of a total of 30 landmarks. Each landmark is a specific region of the face, represented by a number that indicates the location of that region. The landmarks 107, 66, 69, 109, 10, 338, 299, 296, 336, and 9 were extracted from the forehead, the landmarks 118, 119, 100, 126, 209, 49, 129, 203, 205, and 50 were extracted from the left cheek, and the landmarks 347, 348, 329, 355, 429, 279, 358, 423, 425, and 280 were extracted from the right cheek. Every landmark was composed of $30 \times 30$ pixels, and the average across the red, green, and blue (RGB) channels was computed for every landmark. The numbers of the landmarks of each area represent approximately evenly spaced regions of that area.

After all the landmarks were extracted, the RGB signals of each landmark were used as input for the algorithms CHROM, LGI-PPGI, POS, and ICA. These algorithms were chosen because of their effectiveness in separating the color information related to blood flow from the color information not related to blood flow, as well as their ability to extract PPG signals from facial videos. CHROM separates the color information by projecting it onto a set of basis vectors, while LGI-PPGI uses local gradient information to extract PPG signals. POS uses a multichannel blind source separation algorithm to extract signals from different sources, and ICA separates the PPG signals from the other sources of variation in the video. These methods were chosen based on their performance in previous studies and their ability to extract high-quality PPG signals from facial videos[20,23].

### Data processing

For the data processing, the signals used as rPPG are the outputs of the algorithms ICA, CHROM, LGI, and POS, and the cPPG signals were resampled to the same fps as the rPPG. First, the filters detrend and bandpass were applied to both the rPPG and cPPG signals. Bandpass is a sixth-order Butterworth with a cutoff frequency of 0.65–4 Hz. The chosen frequency range was intended to filter out noise in both low and high frequencies. Next, the rPPG signals were filtered by removing low variance signals and were segmented into non-overlapping windows of 10 seconds, followed by min–max normalization. We applied histogram equalization to the obtained spatiotemporal maps, showing a general improvement in the performance of the methods.

### Frequency domain

Spectral analysis was performed on both the rPPG and cPPG signals by applying Welch's method to each window of the constructed rPPG and cPPG signals. The highest peak in the frequency domain was selected as the estimated HR, with alternative methods such as autocorrelation also tested. However, these methods showed minimal absolute differences in beats-per-minute absolute difference ($|\Delta HR|$). Welch's method was deemed useful as it allowed for heart rate evaluation in the frequency domain and demonstrated the predictive capability of each channel's rPPG signal.

### Proposed rPPG construction

The model was trained using data sourced from the PURE dataset. The input data contains information from 10 participants. Each participant was captured across 6 distinct videos, engaging in activities categorized as Steady, Talk, Slow Translation, Fast Translation, Small Rotation, and Medium Rotation. This accounts for a total of 60 videos, with an approximate average duration of 1 min. Each video was transformed to RGB signals. Then, every RGB set of signals representing a video was subdivided into 10-s fragments, with each fragment serving as a unit for training data. The dataset used to train the model contains a total of 339 such samples.

Because the duration of each video is 10 seconds and the frame rate is 30, each sample is represented by three RGB signals composed of 300 time-steps. The RGB signals, serving as training inputs, underwent a transformation process resulting in the derivation of four distinct signals through the application of the POS, CHROM, LGI, and ICA methods. Consequently, each 10-s segment yielded four transformed signals, which were intended for subsequent utilization as input for the model. Before being fed to the model, data preprocessing was applied to the signals. Then, a 5-fold cross-validation (CV) procedure was conducted. During this procedure, the dataset was partitioned into five subsets, with a distribution ratio of 80% for training data and 20% for testing data within each fold.

The model's architecture was composed of four blocks of LSTM and dropout, followed by a dense layer. The model architecture is shown in Fig. 2. To reduce the number of features of the model in each layer, the number of cells in each block decreases from 90 to 1. The learning rate scheduler implemented was ReduceLROnPlateau and the optimizer was Adam[30]. Finally, the metrics root mean squared error (RMSE) and Pearson correlation coefficient ($r$) were set as loss function.

### Evaluation of the signals

To evaluate the signals, we applied four criteria: Dynamic Time Warping (DTW), Pearson's $r$ correlation coefficient, RMSE, and $|\Delta HR|$. We

**Fig. 2 | Architecture of the model.** The model architecture generates a remote photo-plethysmogram (rPPG) signal from three regions of interest: the forehead (R1), left cheek (R2), and right cheek (R3). The average value from each region is calculated, and these averages are then combined to produce the overall rPPG signal. The model is composed of four blocks of LSTM and dropout, followed by a dense layer. The methods ICA, LGI, CHROM, and POS were used as input to the model. rPPG remote photoplethysmogram, RGB red, green, and blue channels, LSTM long short-term memory.

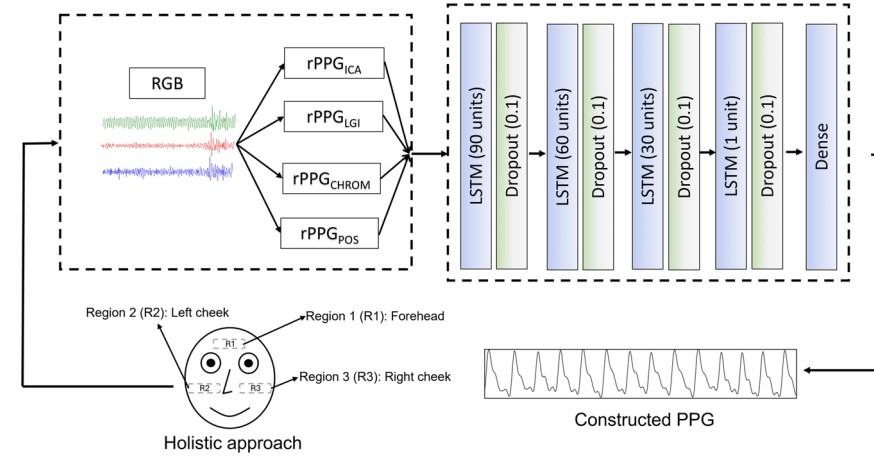

**Fig. 3 | Results of the different methods for the datasets PURE, LGI-PPGI, and MR-NIRP.** Box-plot of the $r$ coefficient and DTW for the different methods. The $p$-value of every method against our model is given above the boxes. The $p$-values are obtained by applying the Friedman and post hoc Nemenyi tests. Taking into account the Bonferroni correction, the adjusted significance level is 0.05/15 = 0.003 given that there are six groups, and a total of 15 tests were performed per dataset and metric. The total number of samples is 339 for PURE, 251 for LGI-PPGI, and 187 for MR-NIRP. The error bar represents 95% confidence interval. **a** Results for the dataset PURE. **b** Results for the dataset LGI-PPGI. **c** Results for the dataset MR-NIRP. $r$, Pearson's correlation coefficient. DTW Dynamic Time Warping. Colors: the purple, blue, and green box colors refer to the results of the datasets PURE, LGI-PPGI, and MR-NIRP, respectively.

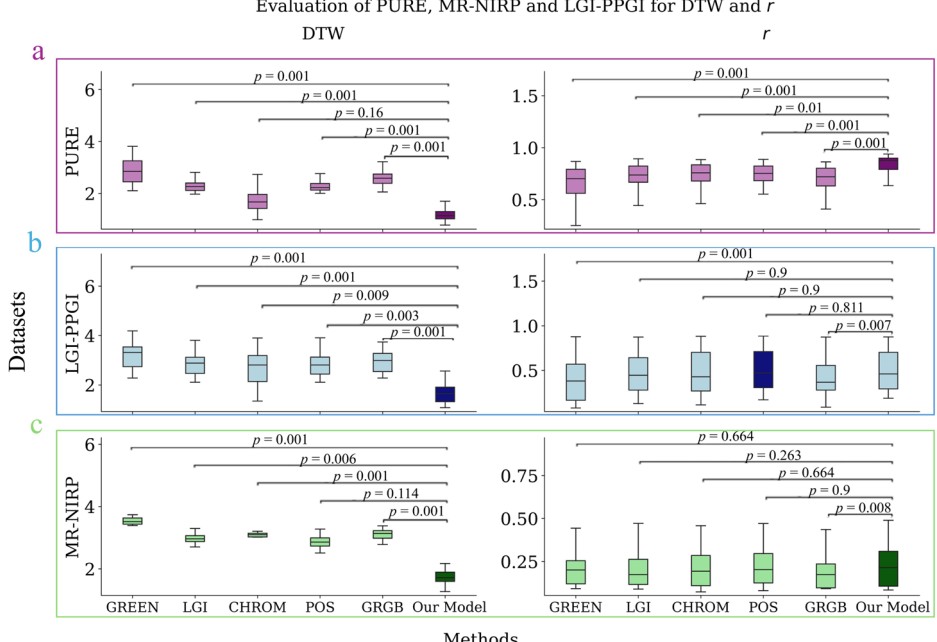

computed each criterion for every window in each video. We then took the average of the values of all the windows to obtain the final results. This helped us to analyze the results of every model from different points of view.

**DTW**. DTW[31] is a useful algorithm for measuring the similarity between two time series, especially when they have varying speeds and lengths. The use of DTW is also relevant for this case because the rPPG and its ground truth may not be aligned sometimes, so metrics that rely on matching timestamps are less appropriate. The metric was implemented using the Python package DTAIDistance[32].

**Pearson's correlation coefficient (r).** The equation below shows how the $r$ coefficient calculates the strength of the relationship between rPPG and cPPG.

$$r = \frac{\sum_{i=1}^{N}(x_i - \hat{x})(y_i - \hat{y})}{\sqrt{\sum_{i=1}^{N}(x_i - \hat{x})^2}\sqrt{\sum_{i=1}^{N}(y_i - \hat{y})^2}} \qquad (1)$$

In this equation, $x_i$ and $y_i$ are the values of the rPPG and PPG signals at lag $i$, respectively. $\hat{x}$ and $\hat{y}$ are their mean values. $N$ is the number of values in the discrete signals.

**RMSE**. The equation below shows how RMSE calculates the prediction error, which is the difference between the ground truth values and the extracted rPPG signals.

$$RMSE = \sqrt{\frac{\sum_{i=1}^{N}(x_i - y_i)^2}{N}} \qquad (2)$$

In this equation, $N$ is the number of values and $x_i$, $y_i$ are the values of the rPPG and contact PPG signals at lag $i$, respectively.

**|ΔHR|**. HR was estimated using Welch's method, which computes the power spectral density of a signal and finds the highest peak in the frequency domain. The peak was searched within a range of 39–240 beats-per-minute (BPM), which is the expected range of human BPMs. |ΔHR| is obtained as the absolute difference between the HR estimated from rPPG and the HR estimated from cPPG.

**Statistical tests**
To evaluate the model's performance, we applied non-parametric statistical tests, which have fewer assumptions about the data distribution than

**Fig. 4 | Results of the different methods for the across activities.** Boxplot of the *r* coefficient and DTW for the different methods. The *p*-value of every method against our model is given above the boxes. The *p*-values are obtained by applying the Friedman and post hoc Nemenyi tests. Taking into account the Bonferroni correction, the adjusted significance level is 0.003. The total number of samples is 339 for PURE, 251 for LGI-PPGI, and 187 for MR-NIRP. The error bar represents 95% confidence interval. Figures **a**–**e** show the results for the activities Rest, Talk, Translation, Rotation, and Gym. *r*, Pearson's correlation coefficient. DTW Dynamic Time Warping.

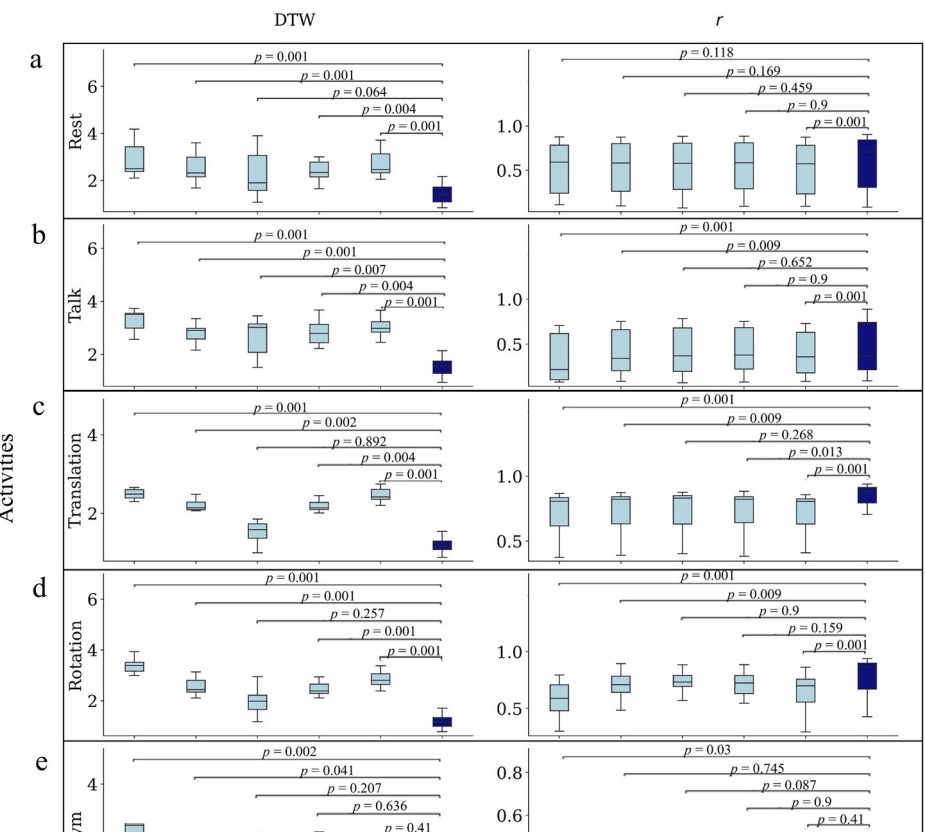

Evaluation of PURE, MR-NIRP and LGI-PPGI for DTW and *r*

parametric ones. Some comparisons involved small sample sizes, such as those with a limited number of subjects.

The Friedman Test[33] is appropriate for this study because it evaluates the means of three or more groups. Every group is represented by a model. If the *p*-value is significant, the means of the groups are not equal. The Nemenyi Test[34] was used to calculate the difference in the average ranking values and then to compare the difference with a critical distance (CD). The general procedure is to apply the Friedmann test to each group and if the *p*-value is significant, the means of the groups are not equal. In that case, the Nemenyi test is performed to compare the methods pairwise. The Nemenyi test helps to identify which methods are similar or different in terms of their average ranks. The Bonferroni correction was applied for multiple-comparison correction.

## Results

To evaluate the performance of the proposed model, we conduct several experiments. The metrics DTW, *r*, RMSE, and |ΔHR| are implemented, and the evaluation is done across datasets and activities.

### Evaluation across datasets

For every dataset, the rPPG obtained from the model is compared to the rPPG obtained from different algorithms. This is done by comparison against the reference PPG, as shown in Fig. 3. The other methods used for comparison are LGI, CHROM, POS, and green-red-green-blue (GRGB)[12]. The green channel is also included as a baseline. Only the dataset PURE is used for training, with a 5-fold CV.

For the metric DTW, the proposed model shows a clear improvement over the other methods. For this metric, CHROM and POS are the best alternatives to our model, but our model shows a significant improvement over POS and CHROM in most cases. Similarly, for *r*, our model outperforms every other method for the PURE dataset. However, even though our model shows the best *r* for the datasets MR-NIRP and LGI-PPGI, the difference is not significant when compared to other models such as CHROM or POS. This is because our model was trained only with data from PURE; however, we can see that it still outperforms the other models in terms of DTW. The results for every metric are shown in Supplementary Table 1, and the *p*-values are shown in Supplementary Table 2.

### Evaluation across activities

The next experiment involves a comparison of performance across activities. The activities are Rest, Talk, Rotation, Gym, and Translation. Rotation takes into account the activity Rotation from LGI-PPGI and the activities Small and Medium Rotation from PURE. Talk includes the activity Talk from LGI-PPGI, PURE, and Motion from MR-NIRP (during the activity Motion, the subjects talk). Lastly, Translation includes the activities Small and Medium Translation from PURE and Gym from LGI-PPGI.

As shown in Fig. 4, the same pattern is repeated in the results for the metrics. For DTW, our model always demonstrates the best performance, for every activity. The best performance occurs in activities where the subject is in a steady position, as well as in activities with different types of movement. This indicates that our model improves the quality of the signal in several scenarios. In the case of the *r* coefficient, our model is always the best

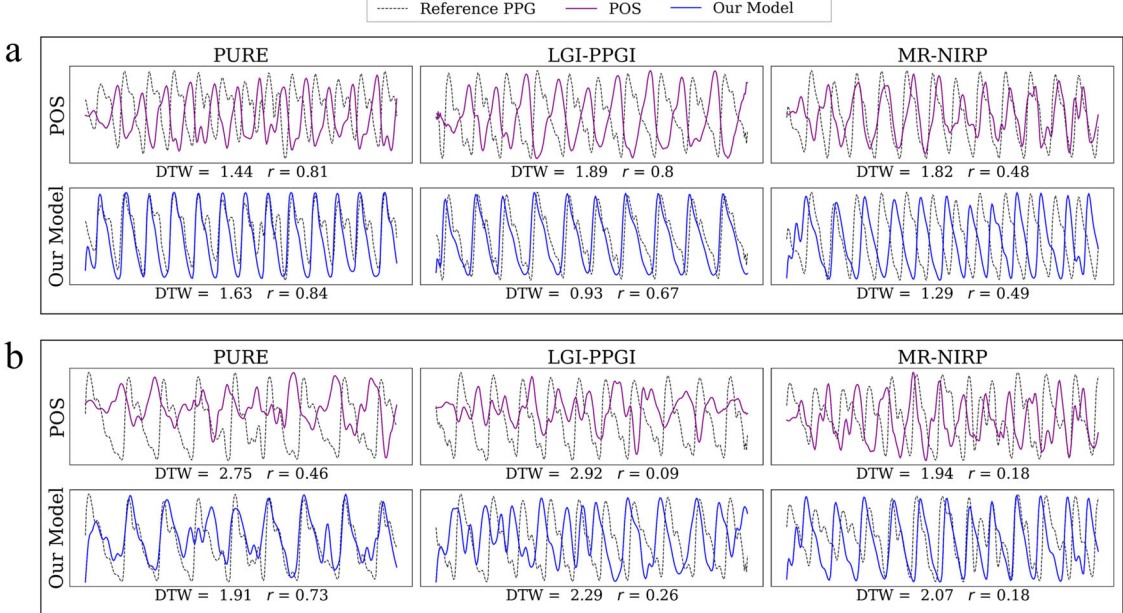

**Fig. 5 | Photoplethysmogram and remote photoplethysmogram extracts from the datasets PURE, LGI-PPGI, and MR-NIRP.** Comparison of our model (blue line) and POS (purple line) against the reference cPPG (black dashed line). **a** Examples are shown where values of the metrics DTW and $r$ are acceptable, for the activity Rest. **b** Extracts from the activity Talk are shown, where the noise is higher in both the rPPG and cPPG signals. $r$, Pearson's correlation coefficient. DTW Dynamic Time Warping, PPG photoplethysmogram. Colors: purple refers to the signal from the model POS, blue to the signal from our model, and black (dashed lines) to the referencce PPG.

performing, except in the case of Gym, where POS outperforms our model. The performance improvement is clearer in terms of DTW, where CHROM is the best alternative; this is in contrast with $r$, where POS is the best alternative. These differences are the reason why comparing different metrics is important. Information about RMSE is included in Supplementary Table 3, and the $p$-values in Supplementary Table 4. The RMSE results are similar to those of DTW: on average the results are better for CHROM, but not significantly. The RMSE of our model is higher in MR-NIRP because in some cases the signals were not aligned, but the morphology is still better than in the other models, as confirmed by DTW.

### Evaluation in the frequency domain
The performance in the frequency domain is only significant when compared to the green channel. Nevertheless, in the case of PURE, the performance of our model is better on average than the rest of the models, with a |ΔHR| of 0.52. For LGI-PPGI, the best results are for POS and our model, with a non-significant difference. In that case, the |ΔHR| of POS is 5.07 and the |ΔHR| of our model is 6.15, which is non-significant, but the difference with the baseline method GREEN is very high, having GREEN a |ΔHR| of 16.09. For the MR-NIRP dataset, our model performs the best, with a |ΔHR| of 7.45. All the results are shown in Supplementary Table 1, and the $p$-values are shown in Supplementary Table 2.

### Overall performance
In order to understand why our model performs well, even for MR-NIRP, where the RMSE of our model is higher than the rest of the models (Supplementary Table 1), it is important to look at the morphology of the signals. In Fig. 5, a comparison between our model and POS is shown for different cases with poor and acceptable results in the metrics. Even though the RMSE of our model is higher in MR-NIRP, we can see that the signal predicted is more robust and less noisy. This is in agreement with the results obtained for the metrics DTW and $r$. The RMSE is higher because the signals are not aligned, leading to an underestimation of the performance of our model. This is why we also included DTW as a metric, given that it helps by aligning these sequences optimally before comparing them, allowing for a fair

comparison even if there are shifts or stretches in the timing of events within the sequences.

## Discussion
Our purpose was to create a model that could improve the quality and robustness of the rPPG signal. Previous studies have mostly focused on improving one physiological parameter derived from the rPPG (e.g., HR or SpO$_2$). By improving the quality of only one physiological parameter, information from the rPPG signal can be lost. Due to this concern, we decided to construct an rPPG signal by making it as similar as possible to the reference PPG. To achieve this, in contrast to other studies, we implemented RMSE, DTW, and $r$ as performance metrics. In most studies[18,35,36], these metrics are calculated from predicted HR and ground truth HR, but these studies do not take into account the morphology of the rPPG, only the most important frequency. Using our evaluation metrics, we are able to take into account the morphology of the wave, that is, all the frequency spectrum rather than a particular frequency.

Through the gathered information and results shown, our model outperforms the other methods in most of the settings. When it comes to the performance across activities, our model had the best performance for the six activities and both metrics DTW and $r$, with the exemption of the activity Gym for the metric $r$, where POS is the best model. In the analysis of performance across datasets, our model shows the lowest DTW for every dataset. It also outperforms other methods for the metrics $r$ and RMSE, except for the MR-NIRP dataset, where the advantage is not always significant. It is important to take a closer look at the results, given that DTW is more informative than RMSE in this case. For RMSE, the distance between each point of two signals is measured, as opposed to DTW, where the speed of the signals is also taken into account. For this study, there are many cases in the predictions of our model where the speed of the signals is slightly different, but the morphology is still similar and informative. The disagreement between the results for DTW and RMSE in the MR-NIRP dataset reflects this, i.e., the morphology of the predicted rPPG signal is acceptable (low DTW), even though is not well aligned with the cPPG (high RMSE). One potential resolution could involve signal alignment through techniques

like cross-correlation, followed by RMSE calculation. This alignment procedure could alleviate the current inconsistency and provide a more unified perspective on signal evaluation. Further exploration and validation of this alignment strategy could shed light on its effectiveness in resolving the observed disagreement between DTW and RMSE. Taking a closer look at the signal, it is apparent that the best alternatives to our model, POS and CHROM, have high variability and a great amount of noise, especially in activities such as Gym or Talk. Furthermore, the prediction of the HR of POS and CHROM is worse than our model, especially in the cases of the datasets PURE and MR-NIRP. In the case of the signal given by our model, the robustness is much better, and even though there might be a slight offset, our model is much closer to the reference PPG in terms of morphology.

In terms of activities, our model has shown great robustness to external noise and movements of the subject. While it outperforms the rest of the methods in simple activities such as resting, the results are still consistent even with more complex activities, such as Rotation and Talk, where the subject moves the most. We have demonstrated that our model achieves an r coefficient up to 0.84 and 0.77 in Translation and Rotation, respectively, which means that our model is able to achieve the best results in many different activities for all three datasets. When it comes to the frequency domain, our model obtained the best results in terms of |ΔHR| for the dataset PURE and MR-NIRP. We can see, especially in PURE, how our model can achieve such a similarity between rPPG and cPPG, with an |ΔHR| of 0.52. For all the activities, the two best models were POS and our model in all the cases.

When contrasting the outcomes of this study with those of others, we find different scenarios. In some studies, the goal was to improve the rPPG wave signal; for example, in ref. 19, the authors used an ML model composed of dense layers to restore the rPPG signal. However, the authors did this for only one dataset composed of five subjects, whereas we did it for a dataset composed of 10 subjects and also tested it in two out-of-distribution datasets, proving improvement in performance in unseen data obtained in different conditions. Additionally, we took into account the distinction between activities and analyzed this distinction, as well as the HR. Another study[18] also trained a model with PURE and obtained an r coefficient of 0.83, the same as in our study. However, the metrics mean absolute error and RMSE that they implemented could not be compared to ours, because in our case the windows of the signals are normalized. In that study, the researchers performed a cross-dataset estimation with another dataset, but did not show the results for the r coefficient. In our study, we performed a further analysis by including two more datasets that the model had not seen, and also evaluated the absolute difference in HR estimation between the rPPG and the reference cPPG.

While most previous studies[35,37] have focused on HR, as discussed above, when a model is created only to detect HR, information about other frequencies and features of the wave is lost, such as the diastolic peak. For the PURE dataset, the results of ETA-rPPG[35] in HR estimation are 0.34 |ΔHR|, which is a better result than ours (0.52). For Siamese-rPPG[18], the |ΔHR| is 0.51, for DeepPhys[38] 0.83, and for HR-CNN[39] 1.84. Nonetheless, we consider it important to mention that our model did not have as a loss function any objective related to HR estimation. Our study shows a model that is able to detect HR, but also potentially other physiological parameters.

As future work, we believe the next steps to follow are the assessment of other physiological parameters, such as beat-to-beat HR assessment and oxygen saturation. This could be useful to identify which physiological parameters can be reliably estimated by the model. Another focal point is to estimate the robustness of the signal produced by the model in different conditions, like the lighting settings or the recording device.

## Conclusion

The results of our proposed model are promising in comparison with the best-performing traditional methods, including POS, LGI, and CHROM and warrant more research. This is confirmed by the metrics DTW, r, and |ΔHR|. We train a model with data only from PURE, and test it on PURE, LGI-PPGI, and MR-NIRP. While the outcome is favorable to our model for the dataset PURE, it is also favorable for the other datasets that were not used in the training. The rPPG estimated from our model is more robust and reliable than other methods for activities where there is more movement of the subjects or the recording is made outdoors. We have successfully developed a method that can construct an rPPG signal that resembles the PPG signal from the fingertip. This offers a contact-free alternative with many future applications, such as Health Monitoring and Telemedicine, and Driver Monitoring and Biometric Authentication.

## Data availability

The PURE[26] dataset is available at https://www.tu-ilmenau.de/universitaet/fakultaeten/fakultaet-informatik-und-automatisierung/profil/institute-und-fachgebiete/institut-fuer-technische-informatik-und-ingenieurinformatik/fachgebiet-neuroinformatik-und-kognitive-robotik/data-sets-code/pulse-rate-detection-dataset-pure. The LGI-PPGI[21] dataset is available at https://github.com/partofthestars/LGI-PPGI-DB. The MR-NIRP[27] dataset is available at https://computationalimaging.rice.edu/mr-nirp-dataset/. The framework pyVHR can be downloaded from https://github.com/phuselab/pyVHR. The source data, obtained in the experiments, is available in the supplementary information.

## Code availability

The code described in this manuscript is publicly accessible on GitHub at https://github.com/rodrigo-castellano/ML_based_rPPG_construction. In addition, the specific version of the code discussed in the paper has been archived and is available via a DOI-minting repository, Zenodo[40].

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

## Acknowledgements

## Author contributions
R.C.O. developed the code, carried out the methodology, experiments, visualization, and wrote the original draft. M.E. designed and led the study. R.C.O., C.M., and M.E. conceived the study. All authors have read and agreed to the published version of the manuscript.

## Funding

## Competing interests
The authors declare no competing interests.
