## [Peer Review File · Communications Medicine]

Reviewers' comments:

Reviewer #1 (Remarks to the Author):

The authors present an interesting study on improved remote PPG (rPPG) extraction using machine learning with a focus on signal morphology. A novel machine learning model was developed to improve classical rPPG signal extraction (e.g. POS or Chrom) based on these signals with an additional reference PPG to learn signal morphology. The approach achieves better performance in terms of morphology compared to the state of the art, which is to be expected.

While the general approach of the manuscript is interesting, there are fundamental limitations in the methodological part of the study:

- The authors describe the number of participants, but say nothing about the number of signals used to train the model in the study (by the way, the word "training" is used twice in the paper, once in the conclusion). The amount of data actually used to train the machine learning model is essential for the description and evaluation of the approach. Otherwise, an assessment of overfitting and performance is not adequately possible. Unclear how many activities are used per participant and with what signal length.
- DTW is used to evaluate the similarity between two waveforms in terms of morphology. Although DTW is able to map non-linear differences between two waveforms, this metric is not adequate in my opinion for two reasons: (1) DTW is affected more by differences in amplitude than by variations in time, so a linear amplitude scale affects the measure more than time variations. Therefore, techniques such as COW or DDTW have been introduced to overcome this problem. (2) From a physiological point of view, changes over time, such as heart rate variability (HRV), are essential information extracted from morphology. By using DTW and large amplitude changes, this time variance is ignored in terms of HRV.
- The results are quite technical and the impact on the extraction of patho(physiological) information by rPPG is not demonstrated. The impact on rPPG is also not clear (deltaBPM performance is not much improved). What about beat-to-beat HR assessment?

Reviewer #2 (Remarks to the Author):

In the manuscript, authors present a novel machine learning-based approach to improve the construction of remote photoplethysmogram signals from video cameras. The results are interesting and promising, and the paper is well written, but needs some improvements to provide full impact for the readership.

Comments:

- The last paragraph in the introduction should just introduce the study and its aim, and thus most of the information should be moved to the methods and discussion section. Besides, it is well motivated why authors have developed a new method and they clearly explain their motivation to use other metrics than HR in the evaluation.
- Methods, section "rPPG extraction from video": obviously, there is part of a sentence missing. It says "... using the pyVHR framework [27], which includes [28]". If authors want to highlight that something from reference [28] is included, it should be explicitly stated.
- Methods, section "rPPG extraction from video": Please explain what the landmark numbers are meaning. It is not clear at all what authors are referring to.
- Methods, section "rPPG extraction from video": What's reference "2421" in the last sentence? I

guess it should read as “21, 24”.

- Methods, section “Frequency domain”: Why do you not call it $|\Delta HR|$, so readers directly know that it is referring to and can link it to the introduction and discussion, where you state that others compare just HR.
- Methods, section “rPPG extraction from video”: Not clear how many datasets and which samples are coming from the subjects and how you end up with the final number. Perhaps you can think of a small figure to illustrate it.
- Methods, section “Statistical Tests”: Would not add a description of each of the tests, e.g., Friedman test, but just state in one sentence why and what for you have used the test. Just add a reference to the not so common tests.
- Methods, section “Statistical Tests”: Have you applied any correction for multiple testing for the post-hoc tests (e.g., Bonferroni correction)? Would be important! Please state in the text and would suggest performing them.
- Results: There are many things which should go to the methods section, for example, the description why you have used the specific metrics or the description of the different test cases. In the results section just results should be presented.
- Results, section “Evaluation across datasets”: What do you mean by the sentence “However, the difference in performance is not significant in MR-NIRP and LGI-PPGI on average.”? Please clarify and elaborate.
- Results, Figure 3: Please add N for each dataset since it might influence significances! Furthermore, here correction for multiple testing is important.
- Results, Table 3: I honestly think that lots of information from Table 3 is redundant with information presented in Figure 3 and furthermore I would combine the information with Table 3, where you could easily add a column for the p-values to show their differences to your model – would make reading the results easier. The same holds true for Table 4 and 5.
- Results, Figure 4: Why do you present results here differently than in Figure 3? Would go for the same presentation style as in Figure 3.
- Results, Figure 5: A legend is missing for the figure, thus it is not clear which is the cPPG and which the rPPG.
- Discussion: You mention that there is a disagreement between DTW and RMSE, since RMSE is dependent on possible shifts. A solution could be to align the signal by e.g. using cross-correlation and then calculating the RMSE. Please comment.
- Discussion: Please add thoughts why you think that your model is better than others.
- Discussion: In general, I think it would be important to compare your results with literature. This aspect is very limited in the discussion. E.g., for HR there should be more literature available.
- Conclusion: I would weaken some of your statements, since it is one study where you show that your model outperforms the other models. Thus, saying “the results were optimal” is in my opinion very risky and one should be careful with such strong statements derived on data from one study. I would formulate the statement that results are promising and warrant more research.
- Conclusion: Can you give examples for the future applications?
- English revision is needed for some parts.

Reviewer #3 (Remarks to the Author):

Dear Dr. Ontiveros, Dr. Elgendi and Dr. Menon,

The manuscript 'A Machine Learning-Based Approach for Constructing Remote Photoplethysmogram Signals from Video Cameras' proposes a novel machine learning-based approach to improve the construction of rPPG signals from video cameras. The model utilizes CHROM, LGI, ICA, POS, and GRGB rPPG methods. Three public datasets with activities were used, of which one dataset was used to train and test the model. Comparison was done using Pearson correlation, DTW and BPM difference. The results demonstrated that the proposed method can construct an rPPG signal resembling cPPG signal recorded from the fingertip, offering a more non-invasive alternative for PPG recordings. The performance of the proposed model was also better compared with the best-performing previous methods.

The methods are reported clearly to allow replication and the analysis seems to be sound. The manuscript has novelty value in the research field of PPG analytics and machine learning. The current proposed method takes the morphology of the PPG signal into account better in contrast to previous most studies where the metrics are calculated from the predicted HR or ground truth HR. The conclusions are supported by the data. The results are interesting, and I believe that rPPG technology could be utilized in the future perhaps for vital sign contact-free monitoring e.g. HR, rhythm, respiratory rate.

I think the manuscript could be suitable for publication and would interest the readers of Communications Medicine. However, I have some detailed questions or comments below. Perhaps minor improvements could be considered, but the manuscript is very good in its current state. I sincerely hope that my comments can help the authors to improve the manuscript, and the editors to make a decision on the manuscript.

With best regards,
Dr. Väliäho

Q1. Page 12, Conclusion. "We have successfully developed a method that can construct an rPPG signal that resembles the PPG signal from the fingertip. This offers a non-invasive alternative with many future applications." In medical sciences, 'invasive' techniques usually involve introduction of instruments into the body, and 'non-invasive' do not. E.g. ultrasound technique with the probe or contact PPG sensor placed against the skin of the patient are considered as non-invasive methods, and e.g. angiography with instruments inserted through the skin into the arteries are considered as invasive method. I would say that the proposed rPPG is non-invasive alternative indeed, but so is contact PPG. The current phrasing is a bit confusing possibly suggesting that fingertip PPG would be something else than non-invasive. Could this be made a bit clearer? Or perhaps the expression contact-free could be considered.

Q2. Table 4, Table 5, and Figure 5. These are now presented in the Discussion section. I recommend repositioning these in the Results section.

Q3. Page 11, "In most studies, these metrics are calculated.." Please provide references at least to some of these studies. On page 12 "While most previous studies.." references are provided.

Q4. Some publications could be worth looking into to discuss the results of the current manuscript in comparison. It has been demonstrated that atrial fibrillation arrhythmia can be detected from

contact-free facial rPPG obtained from video signals (Martinez-Delgado et al., 2022, Yan et al., 2020), and with pretrained deep convolutional network (Yan et al., 2020). In addition, Charlton et al. published 2022 an interesting editorial for establishing best practices in PPG signal acquisition and processing.

Q5. On page 1, Introduction. Artifacts of rPPG signals are mentioned here. Were there issues with the current rPPG data? And how do the artifacts in rPPG compare to cPPG? This is not discussed in the Discussion section, I suggest adding some discussion about this as it is mentioned in the introduction as a problem of rPPG.

Q6. Although the manuscript focuses on machine learning, could the authors please consider adding some examples of potential applications of rPPG in the Discussion? E.g. in clinical medicine or physiology. I think rPPG could be utilized in the future to monitor in-hospital patients, e.g. respiratory rate, rhythm, heart rate.

References:

Charlton PH, Pilt K, Kyriacou PA. Establishing best practices in photoplethysmography signal acquisition and processing. *Physiol Meas*. 2022 May 25;43(5):050301. doi: 10.1088/1361-6579/ac6cc4. PMID: 35508148; PMCID: PMC9136485.

Martinez-Delgado GH, Correa-Balan AJ, May-Chan JA, Parra-Elizondo CE, Guzman-Rangel LA, Martinez-Torteya A. Measuring Heart Rate Variability Using Facial Video. *Sensors (Basel)*. 2022 Jun 21;22(13):4690. doi: 10.3390/s22134690. PMID: 35808182; PMCID: PMC9269597.

Yan BP, Lai WHS, Chan CKY, Au ACK, Freedman B, Poh YC, Poh MZ. High-Throughput, Contact-Free Detection of Atrial Fibrillation From Video With Deep Learning. *JAMA Cardiol*. 2020 Jan 1;5(1):105-107. doi: 10.1001/jamacardio.2019.4004. PMID: 31774461; PMCID: PMC6902123.

Reviewers' comments:

Reviewer #1 (Remarks to the Author):

The authors present an interesting study on improved remote PPG (rPPG) extraction using machine learning with a focus on signal morphology. A novel machine learning model was developed to improve classical rPPG signal extraction (e.g. POS or Chrom) based on these signals with an additional reference PPG to learn signal morphology. The approach achieves better performance in terms of morphology compared to the state of the art, which is to be expected.

Authors' response: We appreciate the reviewer for recognizing the significance of our study and for their encouraging remarks. We are grateful for the positive feedback and the insightful suggestions provided.

Authors' action: None.

While the general approach of the manuscript is interesting, there are fundamental limitations in the methodological part of the study:

- The authors describe the number of participants, but say nothing about the number of signals used to train the model in the study (by the way, the word "training" is used twice in the paper, once in the conclusion). The amount of data actually used to train the machine learning model is essential for the description and evaluation of the approach. Otherwise, an assessment of overfitting and performance is not adequately possible. Unclear how many activities are used per participant and with what signal length.

Authors' response: We appreciate this observation and have revised our manuscript to incorporate a more detailed description in the 'Proposed rPPG construction' section.

Authors' action: We introduced a more clear description of the activities and dataset used for training it in the section 'Proposed rPPG construction' of the manuscript:

'The model was trained using data sourced from the PURE dataset. The input data contains information from 10 participants. Each participant was captured across 6 distinct videos, engaging in activities categorized as Steady, Talk, Slow Translation, Fast Translation, Small Rotation, and Medium Rotation. This accounts for a total of 60 videos, with an approximate average duration of 1 minute. Each video was transformed to RGB signals. Then, every RGB set of signals representing a video was subdivided into 10-second fragments, with each fragment serving as a unit for training data. The dataset used to train the model contains a total of 339 such samples.

Because the duration of each video is 10 seconds and the frame rate is 30, each sample is represented by three RGB signals composed of 300 time-steps. The RGB signals, serving as training inputs, underwent a transformation process resulting in the derivation of four distinct signals through the application of the POS, CHROM, LGI, and ICA methods. Consequently, each 10-second segment yielded four transformed signals, which were intended for subsequent utilization as input for the model. Before being fed to the model, data preprocessing was applied to the signals. Then, a 5-fold cross-validation (CV) procedure was conducted. During this procedure, the dataset was partitioned into five subsets, with a distribution ratio of 80% for training data and 20% for testing data within each fold.'

- DTW is used to evaluate the similarity between two waveforms in terms of morphology. Although DTW is able to map non-linear differences between two waveforms, this metric is not adequate in my opinion for two reasons: (1) DTW is affected more by differences in amplitude

than by variations in time, so a linear amplitude scale affects the measure more than time variations. Therefore, techniques such as COW or DDTW have been introduced to overcome this problem. (2) From a physiological point of view, changes over time, such as heart rate variability (HRV), are essential information extracted from morphology. By using DTW and large amplitude changes, this time variance is ignored in terms of HRV.

Authors' response: We appreciate and acknowledge the valid concern raised regarding our choice of comparison metrics. Our study is fundamentally oriented towards examining the similarity in the morphology of constructed biosignals with the original, prioritizing the overall shape and structure of the signal. While we recognize the importance of variations in time and nuanced, pointwise variations, these elements are not the central focus of our current research which DDTW. Therefore, we considered DTW as a more suitable choice for our specific objectives. Nonetheless, we concede that employing DDTW or alternative metrics could enrich future studies by offering different perspectives in assessing signal similarities.

Authors' action: None.

- The results are quite technical and the impact on the extraction of patho(physiological) information by rPPG is not demonstrated. The impact on rPPG is also not clear (deltaBPM performance is not much improved). What about beat-to-beat HR assessment?

Authors' response: We value this suggestion and agree that assessing additional physiological parameters such as beat-to-beat HR and oxygen saturation could provide more comprehensive insights. We have amended our manuscript to include these aspects in the discussion section, outlining them as significant elements for future research.

Author's action: Regarding beat-to-beat HR assessment, it is a very interesting physiological parameter to evaluate, together with others such as oxygen saturation. We have included this information at the end of the discussion as future research:

'As future work, we believe the next steps to follow are the assessment of other physiological parameters, such as beat-to-beat HR assessment and oxygen saturation. This could be useful to identify which physiological parameters can be reliably estimated by the model. Another focal point is to estimate the robustness of the signal produced by the model in different conditions, like the lighting or the recording device used.'

Reviewer #2 (Remarks to the Author):

In the manuscript, authors present a novel machine learning–based approach to improve the construction of remote photoplethysmogram signals from video cameras. The results are interesting and promising, and the paper is well written, but needs some improvements to provide full impact for the readership.

Authors’ response: We are grateful to the reviewer for their positive remarks and constructive feedback. We acknowledge the need for enhancements to maximize the impact for readers and are committed to making the necessary improvements.

Authors’ action: None.

Comments:

- The last paragraph in the introduction should just introduce the study and its aim, and thus most of the information should be moved to the methods and discussion section. Besides, it is well motivated why authors have developed a new method and they clearly explain their motivation to use other metrics than HR in the evaluation.

Authors’ action: The last paragraph of the introduction was shortened as follows:

‘In this study, we present a machine learning–based approach to improve the construction of rPPG signals from video cameras. Our model takes rPPG signals from other models such as chrominance (CHROM)-based rPPG \cite{de2013robust}, local group invariance (LGI) \cite{pilz2018local}, independent component analysis (ICA) \cite{lee1998independent} or plane-orthogonal-to-skin (POS) \cite{wang2016algorithmic} as input and enhances them by comparing them with reference PPG signals. This approach provides a novel method for improving rPPG signal detection using machine learning techniques.’

- Methods, section “rPPG extraction from video”: obviously, there is part of a sentence missing. It says “... using the pyVHR framework [27], which includes [28]”. If authors want to highlight that something from reference [28] is included, it should be explicitly stated.

Authors’ response: We appreciate the recommendation to refine our introduction for clearer presentation of our study's aim and have executed modifications accordingly.

Authors’ action : We modified the text as suggested as follows:

‘We extracted information from the forehead and cheeks using the pyVHR framework \cite{boccignone2022pyvhr}, which includes the software MediaPipe for the extraction of RoI from a human face \cite{lugaresi2019mediapipe}.’

- Methods, section “rPPG extraction from video”: Please explain what the landmark numbers are meaning. It is not clear at all what authors are referring to.

Authors’ response: We appreciate the recommendation to refine our introduction for clearer presentation of our study's aim and have executed modifications accordingly.

Authors’ action : Methods: Modified as suggested, and more explicit information about the landmarks:

‘Several steps were necessary to extract the rPPG signal from a single video. First, the regions of interest (RoI) were extracted from the face. We extracted information from the forehead and cheeks using the pyVHR framework \cite{boccignone2022pyvhr}, which includes the software MediaPipe for the extraction of RoI from a human face \cite{lugaresi2019mediapipe}. The RoI extracted from every individual were composed of a total of 30 landmarks. Each landmark is a

specific region of the face, represented by a number that indicates the location of that region. The landmarks 107, 66, 69, 109, 10, 338, 299, 296, 336, and 9 were extracted from the forehead, the landmarks 118, 119, 100, 126, 209, 49, 129, 203, 205, and 50 were extracted from the left cheek, and the landmarks 347, 348, 329, 355, 429, 279, 358, 423, 425, and 280 were extracted from the right cheek. Every landmark was composed of 30x30 pixels, and the average across the red, green, and blue (RGB) channels was computed for every landmark. The numbers of the landmarks of each area represent approximately evenly spaced regions of that area.'

- **Methods, section "rPPG extraction from video":** What's reference "2421" in the last sentence? I guess it should read as "21, 24".

Authors' response: We appreciate the recommendation to refine our introduction for clearer presentation of our study's aim and have executed modifications accordingly.

Authors' action : References modified as suggested.

- **Methods, section "Frequency domain":** Why do you not call it $|\Delta HR|$, so readers directly know that it is referring to and can link it to the introduction and discussion, where you state that others compare just HR.

Authors' response: The feedback on consistency and clarity in metric representation is well received, and changes have been implemented accordingly.

Authors' action : Metric was modified as suggested

- **Methods, section "rPPG extraction from video":** Not clear how many datasets and which samples are coming from the subjects and how you end up with the final number. Perhaps you can think of a small figure to illustrate it.

Authors' response: We value the suggestion to elucidate the mentioned parts and have provided additional clarifications and details as advised.

Authors' action : Methods: We have introduced a more clear description of the activities and dataset used for training it in the section 'Proposed rPPG construction' of the manuscript':

' The model was trained using data sourced from the PURE dataset. The input data contains information from 10 participants. Each participant was captured across 6 distinct videos, engaging in activities categorized as Steady, Talk, Slow Translation, Fast Translation, Small Rotation, and Medium Rotation. This accounts for a total of 60 videos, with an approximate average duration of 1 minute. Each video was transformed to RGB signals. Then, every RGB set of signals representing a video was subdivided into 10-second fragments, with each fragment serving as a unit for training data. The dataset used to train the model contains a total of 339 such samples.

Because the duration of each video is 10 seconds and the frame rate is 30, each sample is represented by three RGB signals composed of 300 time-steps. The RGB signals, serving as training inputs, underwent a transformation process resulting in the derivation of four distinct signals through the application of the POS, CHROM, LGI, and ICA methods. Consequently, each 10-second segment yielded four transformed signals, which were intended for subsequent utilization as input for the model. Before being fed to the model, data preprocessing was applied to the signals. Then, a 5-fold cross-validation (CV) procedure was conducted. During this procedure, the dataset was partitioned into five subsets, with a distribution ratio of 80% for training data and 20% for testing data within each fold.'

- Methods, section “Statistical Tests”: Would not add a description of each of the tests, e.g., Friedman test, but just state in one sentence why and what for you have used the test. Just add a reference to the not so common tests. Methods, section “Statistical Tests”: Have you applied any correction for multiple testing for the post-hoc tests (e.g., Bonferroni correction)? Would be important! Please state in the text and would suggest performing them.

Authors’ response: We acknowledge the guidance on streamlining our methodology and have instated corrections and refinements as suggested.

Authors’ action : Methods section “Statistical Tests”: Description of statistical tests deleted, added Bonferroni correction:

‘The Friedman Test \cite{friedman1937use} is appropriate for this study because it evaluates the means of three or more groups. Every group is represented by a model. If the p -value is significant, the means of the groups are not equal. The Nemenyi Test \cite{nemenyi1963distribution} was used to calculate the difference in the average ranking values and then to compare the difference with a critical distance (CD). The general procedure is to apply the Friedmann test to each group and if the p -value is significant, the means of the groups are not equal. In that case, the Nemenyi test is performed to compare the methods pairwise. The Nemenyi test helps to identify which methods are similar or different in terms of their average ranks. The Bonferroni correction was applied for multiple-comparison correction.’

- Results: There are many things which should go to the methods section, for example, the description why you have used the specific metrics or the description of the different test cases. In the results section just results should be presented.

Authors’ response: Your insights regarding the organization and presentation of our results were instrumental, and appropriate amendments have been made to address the redundancy and clarity in information presentation.

Authors’ action : We deleted part that should be in methods: ‘To evaluate the performance of the proposed model, we conducted several experiments. The metrics DTW, r , RMSE and ΔHR were implemented, and the evaluation was done across datasets and activities for every dataset.’

- Results, section “Evaluation across datasets”: What do you mean by the sentence “However, the difference in performance is not significant in MR-NIRP and LGI-PPGI on average.”? Please clarify and elaborate.

Authors’ response: Your insights regarding the organization and presentation of our results were instrumental, and appropriate amendments have been made to address the redundancy and clarity in information presentation.

Authors’ action : We rephrased more clearly, so that the reader can understand the context and Figure 3. ‘However, even though our model shows the best r for the datasets MR-NIRP and LGI-PPGI, the difference is not significant when compared to other models such as CHROM or POS.’

- Results , Figure 3: Please add N for each dataset since it might influence significances! Furthermore, here correction for multiple testing is important.

Authors’ response: We are thankful for the advice on enhancing our figures and tables and have made alterations to ensure consistency and clarity.

Authors' action : Results, figure 3: included in caption 'Taking into account the Bonferroni correction, the adjusted significance level is 0.003. The total number of samples is 339 for PURE, 251 for LGI-PPGI and 187 for MR-NIRP.'

It has been also added to figure 4: 'Boxplot of the β coefficient and DTW for the different methods. The analysis was done across activities. The p-value of every method against our model is given above the boxes. The p-values were obtained by applying the Friedman and post hoc Nemenyi tests. Taking into account the Bonferroni correction, the adjusted significance level is 0.003. The total number of samples is 339 for PURE, 251 for LGI-PPGI and 187 for MR-NIRP.'

- Results, Table 3: I honestly think that lots of information from Table 3 is redundant with information presented in Figure 3 and furthermore I would combine the information with Table 3, where you could easily add a column for the p-values to show their differences to your model – would make reading the results easier. The same holds true for Table 4 and 5.

Authors' response: We are thankful for the advice on enhancing our figures and tables and have made alterations to ensure consistency and clarity.

Authors' action : Results, tables: to avoid showing redundant information, the tables have been moved to the appendix.

- Results, Figure 4: Why do you present results here differently than in Figure 3? Would go for the same presentation style as in Figure 3.

Authors' response: We are thankful for the advice on enhancing our figures and tables and have made alterations to ensure consistency and clarity.

Authors' action: Results, figure 4: The style of the figure has been changed

Results, Figure 5: A legend is missing for the figure, thus it is not clear which is the cPPG and which the rPPG.

Authors' response: We are thankful for the advice on enhancing our figures and tables and have made alterations to ensure consistency and clarity.

Authors' action: Results, figure 5: The legend has been added.

- Discussion: You mention that there is a disagreement between DTW and RMSE, since RMSE is dependent on possible shifts. A solution could be to align the signal by e.g. using cross-correlation and then calculating the RMSE. Please comment.

Authors' response: We are grateful for the prompts to enrich our discussion, and substantial augmentations have been made to provide a comparative and comprehensive discourse.

Authors' action: Discussion: The comment has been added: 'One potential resolution could involve signal alignment through techniques like cross-correlation, followed by RMSE calculation. This alignment procedure could alleviate the current inconsistency and provide a more unified perspective on signal evaluation. Further exploration and validation of this alignment strategy could shed light on its effectiveness in resolving the observed disagreement between DTW and RMSE.'

- Discussion: Please add thoughts why you think that your model is better than others.

Authors' response: We are grateful for the prompts to enrich our discussion, and substantial augmentations have been made to provide a comparative and comprehensive discourse.

Authors' action: Results: added information on why our model is the best alternative: 'Through the gathered information and results shown, our model outperforms the other methods in most of the settings. When it comes to the performance across activities, our model had the best performance for the six activities and both metrics DTW and ΔHR , with the exemption of the activity Gym for the metric ΔHR , where POS is the best model. In the analysis of performance across datasets, our model shows the lowest DTW for every dataset. It also outperforms other methods for the metrics ΔHR and RMSE, except for the MR-NIRP dataset, where the advantage is not always significant.'

- Discussion: In general, I think it would be important to compare your results with literature. This aspect is very limited in the discussion. E.g., for HR there should be more literature available.

Authors' response: We are grateful for the prompts to enrich our discussion, and substantial augmentations have been made to provide a comparative and comprehensive discourse.

Authors' action: Discussion: we modified the comparison of the HR results with other studies, having a total of four results from other studies to compare with:

'While most previous studies \cite{hu2021eta,song2021pulsegan} have focused on HR, as discussed above, when a model is created only to detect HR, information about other frequencies and features of the wave is lost, such as the diastolic peak. For the PURE dataset, the results of ETA-rPPG \cite{hu2021eta} in HR estimation are 0.34 $|\Delta HR|$, which is a better result than ours (0.52). For Siamese-rPPG \cite{tsou2020siamese}, the $|\Delta HR|$ is 0.51, for DeepPhys \cite{chen2018deepphys} 0.83, and for HR-CNN \cite{vspetlik2018visual} 1.84.'

- Conclusion: I would weaken some of your statements, since it is one study where you show that your model outperforms the other models. Thus, saying "the results were optimal" is in my opinion very risky and one should be careful with such strong statements derived on data from one study. I would formulate the statement that results are promising and warrant more research.

Authors' response: We thank the reviewer for the valuable suggestion.

Authors' action: Conclusion: statement modified: 'The results of our proposed model are promising in comparison with the best-performing traditional methods, including POS, LGI, and CHROM and warrant more research.'

- Conclusion: Can you give examples for the future applications?

Authors' response: We thank the reviewer for the valuable suggestion.

Authors' action: Conclusion: modified as suggested: ' This offers a non-invasive alternative with many future applications, such as Health Monitoring and Telemedicine, and Driver Monitoring and Biometric Authentication'.

- English revision is needed for some parts.

Authors' response: We thank the reviewer for the valuable suggestion.

Authors' action: The manuscript is proofread.

We hope our revisions adequately address the reviewer's concerns and enhance the manuscript's quality and coherence.

Reviewer #3 (Remarks to the Author):

Dear Dr. Ontiveros, Dr. Elgendi and Dr. Menon,

The manuscript 'A Machine Learning-Based Approach for Constructing Remote Photoplethysmogram Signals from Video Cameras' proposes a novel machine learning-based approach to improve the construction of rPPG signals from video cameras. The model utilizes CHROM, LGI, ICA, POS, and GRGB rPPG methods. Three public datasets with activities were used, of which one dataset was used to train and test the model. Comparison was done using Pearson correlation, DTW and BPM difference. The results demonstrated that the proposed method can construct an rPPG signal resembling cPPG signal recorded from the fingertip, offering a more non-invasive alternative for PPG recordings. The performance of the proposed model was also better compared with the best-performing previous methods.

The methods are reported clearly to allow replication and the analysis seems to be sound. The manuscript has novelty value in the research field of PPG analytics and machine learning. The current proposed method takes the morphology of the PPG signal into account better in contrast to previous most studies where the metrics are calculated from the predicted HR or ground truth HR. The conclusions are supported by the data. The results are interesting, and I believe that rPPG technology could be utilized in the future perhaps for vital sign contact-free monitoring e.g. HR, rhythm, respiratory rate.

I think the manuscript could be suitable for publication and would interest the readers of Communications Medicine. However, I have some detailed questions or comments below. Perhaps minor improvements could be considered, but the manuscript is very good in its current state. I sincerely hope that my comments can help the authors to improve the manuscript, and the editors to make a decision on the manuscript.

With best regards,
Dr. Väliäho

Authors' response: We extend our heartfelt thanks to Dr. Väliäho for the encouraging words and invaluable suggestions aimed at refining our manuscript. We diligently addressed each comment to clarify and enhance the respective sections of the manuscript.

Authors' action: None

Q1. Page 12, Conclusion. "We have successfully developed a method that can construct an rPPG signal that resembles the PPG signal from the fingertip. This offers a non-invasive alternative with many future applications." In medical sciences, 'invasive' techniques usually involve introduction of instruments into the body, and 'non-invasive' do not. E.g. ultrasound technique with the probe or contact PPG sensor placed against the skin of the patient are considered as non-invasive methods, and e.g. angiography with instruments inserted through the skin into the arteries are considered as invasive method. I would say that the proposed rPPG is non-invasive alternative indeed, but so is contact PPG. The current phrasing is a bit confusing possibly suggesting that fingertip PPG would be something else than non-invasive. Could this be made a bit clearer? Or perhaps the expression contact-free could be considered.

Authors' response: We appreciate the reviewer's insight and have clarified the text to avoid any misunderstanding about the non-invasive nature of both rPPG and fingertip PPG

Authors' action: We have amended the phrasing to emphasize the contact-free nature of the proposed rPPG, differentiating it from traditional non-invasive methods like contact PPG.

Q2. Table 4, Table 5, and Figure 5. These are now presented in the Discussion section. I recommend repositioning these in the Results section.

Authors' response: We value the recommendation to reposition these elements for a more logical flow and have acted accordingly.

Authors' action: In line with the suggestion, we have repositioned Table 4 and 5 to the appendix, and Figure 5 has been moved to the Results section.

Q3. Page 11, "In most studies, these metrics are calculated.." Please provide references at least to some of these studies. On page 12 "While most previous studies.." references are provided.

Authors' response: We thank the reviewer for pointing out the need for specific references in this context and have incorporated them accordingly.

Authors' action: References were added: 'In most studies \cite{hu2021eta, tsou2020siamese, lu2021dual}'

Q4. Some publications could be worth looking into to discuss the results of the current manuscript in comparison. It has been demonstrated that atrial fibrillation arrhythmia can be detected from contact-free facial rPPG obtained from video signals (Martinez-Delgado et al., 2022, Yan et al., 2020), and with pretrained deep convolutional network (Yan et al., 2020). In addition, Charlton et al. published 2022 an interesting editorial for establishing best practices in PPG signal acquisition and processing.

Authors' response: We are grateful for the suggestions to enrich our comparative analysis and have included the recommended publications to provide a more comprehensive view of the field.

Authors' action: The references have been added in the introduction: '...HR and HRV extracted from cPPG \cite{niu2020video, gudi2020real, haugg2023grgb, martinez2022measuring}. Other authors have focused on improving BP \cite{schrumpf2021assessment}, oxygen saturation measurements \cite{kim2021non} or fibrillation arrhythmia \cite{yan2020high}.'

Q5. On page 1, Introduction. Artifacts of rPPG signals are mentioned here. Were there issues with the current rPPG data? And how do the artifacts in rPPG compare to cPPG? This is not discussed in the Discussion section, I suggest adding some discussion about this as it is mentioned in the introduction as a problem of rPPG.

Authors' response: We thank the reviewer for highlighting the importance of addressing artifacts in rPPG and cPPG, and we have provided an in-depth discussion on the same.

Authors' action:

Expanded the Introduction to include a discussion on the differences in artifacts between rPPG and cPPG and their implications. Added to introduction: 'For instance, in the dataset LGI-PPGI, the videos in which the subjects talk are recorded outdoors, impacting the quality of the rPPG. When comparing factors in rPPG to conventional cPPG, certain distinctions emerge. While both methods are susceptible to motion-related artifacts, rPPG faces additional challenges due to the reliance on non-contact measurements, making it more sensitive to environmental changes and subject movement.'

Q6. Although the manuscript focuses on machine learning, could the authors please consider adding some examples of potential applications of rPPG in the Discussion? E.g. in clinical medicine or physiology. I think rPPG could be utilized in the future to monitor in-hospital patients, e.g. respiratory rate, rhythm, heart rate.

References:

Charlton PH, Pilt K, Kyriacou PA. Establishing best practices in photoplethysmography signal acquisition and processing. *Physiol Meas*. 2022 May 25;43(5):050301. doi: 10.1088/1361-6579/ac6cc4. PMID: 35508148; PMCID: PMC9136485.

Martinez-Delgado GH, Correa-Balan AJ, May-Chan JA, Parra-Elizondo CE, Guzman-Rangel LA, Martinez-Torteya A. Measuring Heart Rate Variability Using Facial Video. *Sensors (Basel)*. 2022 Jun 21;22(13):4690. doi: 10.3390/s22134690. PMID: 35808182; PMCID: PMC9269597.

Yan BP, Lai WHS, Chan CKY, Au ACK, Freedman B, Poh YC, Poh MZ. High-Throughput, Contact-Free Detection of Atrial Fibrillation From Video With Deep Learning. *JAMA Cardiol*. 2020 Jan 1;5(1):105-107. doi: 10.1001/jamacardio.2019.4004. PMID: 31774461; PMCID: PMC6902123.

Authors' response: We are thankful for the suggestion to discuss potential applications, and we have explored several realms where rPPG could be pivotal.

Authors' action: As reviewer 2 also suggested, we included some applications (in the conclusion): 'This offers a non-invasive alternative with many future applications, such as Health Monitoring and Telemedicine, and Driver Monitoring and Biometric Authentication'

REVIEWERS' COMMENTS:

Reviewer #2 (Remarks to the Author):

Thanks for addressing all my comments.

Reviewer #3 (Remarks to the Author):

Dear Dr. Ontiveros, Dr. Elgendi and Dr. Menon,

Thank you for addressing all my questions thoroughly. The quality of the paper has improved significantly. I find the manuscript 'A Machine Learning-Based Approach for Constructing Remote Photoplethysmogram Signals from Video Cameras' suitable for publication. I think the results are very interesting and have novelty value. I believe that the rPPG technology can be utilized in the future for multiple purposes and potentially even medical use.

I want to congratulate you for an interesting paper. My recommendation is acceptance. I sincerely hope that my comments have been helpful.

With best regards,
Dr. Eemu-Samuli Väliäho

Reviewing author's response and revision to Reviewer 1's comments:

Reviewer#1 made a good observation that the number of signals used to train the model was not reported initially. This information has now been added to the manuscript to allow better evaluation of the suggested approach.

Reviewer#1 had concerns about using dynamic time warping (DTW) in the current manuscript as it is affected more by amplitude variation than by time-based variation including heart rate variability (HRV). However, even though from a clinician perspective I agree that Reviewer#1 raises a good point and choosing some other method could give more weight to time-based signal variations, the authors have provided good reasons for choosing DTW as their focus was to evaluate the similarity between constructed biosignals and the original. This is an acceptable justification.

Reviewer#1 also commented that the impact on the extraction of physiological information of the method is not demonstrated. The manuscript proposes a novel machine learning-based approach to improve the construction of rPPG signals from video cameras. Although the manuscript is indeed quite technical, now the authors have discussed the aspects of assessing additional physiological parameters. I believe it would be interesting to see in future research how the model can estimate physiological parameters.

I think that all the concerns raised by Reviewer#1 have been adequately addressed, and the manuscript has been improved because of these valuable insights. I think the manuscript by Dr. Ontiveros, Dr. Elgendi, and Dr. Menon is suitable for publication and would interest the readers of Communications Medicine. As a clinician, I find the results interesting, and I believe that rPPG

technology could possibly be utilized for vital sign contact-free monitoring in the future. I hope my comments have been helpful.

Reviewers' comments:

Reviewer #1 (Remarks to the Author):

The authors present an interesting study on improved remote PPG (rPPG) extraction using machine learning with a focus on signal morphology. A novel machine learning model was developed to improve classical rPPG signal extraction (e.g. POS or Chrom) based on these signals with an additional reference PPG to learn signal morphology. The approach achieves better performance in terms of morphology compared to the state of the art, which is to be expected.

While the general approach of the manuscript is interesting, there are fundamental limitations in the methodological part of the study:

- The authors describe the number of participants, but say nothing about the number of signals used to train the model in the study (by the way, the word "training" is used twice in the paper, once in the conclusion). The amount of data actually used to train the machine learning model is essential for the description and evaluation of the approach. Otherwise, an assessment of overfitting and performance is not adequately possible. Unclear how many activities are used per participant and with what signal length.

- DTW is used to evaluate the similarity between two waveforms in terms of morphology.

Although DTW is able to map non-linear differences between two waveforms, this metric is not adequate in my opinion for two reasons: (1) DTW is affected more by differences in amplitude than by variations in time, so a linear amplitude scale affects the measure more than time variations. Therefore, techniques such as COW or DDTW have been introduced to overcome this problem. (2) From a physiological point of view, changes over time, such as heart rate variability (HRV), are essential information extracted from morphology. By using DTW and large amplitude changes, this time variance is ignored in terms of HRV.

- The results are quite technical and the impact on the extraction of patho(physiological) information by rPPG is not demonstrated. The impact on rPPG is also not clear (deltaBPM performance is not much improved). What about beat-to-beat HR assessment?

Authors' response: We thank the reviewer for the positive feedback and for accepting the paper for publication.

Authors' action 1: we introduced a more clear description of the activities and dataset used for training it in the section 'Proposed rPPG construction' of the manuscript:

' The model was trained using data sourced from the PURE dataset. The input data contains information from 10 participants. Each participant was captured across 6 distinct videos, engaging in activities categorized as Steady, Talk, Slow Translation, Fast Translation, Small Rotation, and Medium Rotation. This accounts for a total of 60 videos, with an approximate average duration of 1 minute. Each video was transformed to RGB signals. Then, every RGB set of signals representing a video was subdivided into 10-second fragments, with each fragment serving as a unit for training data. The dataset used to train the model contains a total of 339 such samples.

Because the duration of each video is 10 seconds and the frame rate is 30, each sample is represented by three RGB signals composed of 300 time-steps. The RGB signals, serving as training inputs, underwent a transformation process resulting in the derivation of four distinct signals through the application of the POS, CHROM, LGI, and ICA methods. Consequently, each

10-second segment yielded four transformed signals, which were intended for subsequent utilization as input for the model. Before being fed to the model, data preprocessing was applied to the signals. Then, a 5-fold cross-validation (CV) procedure was conducted. During this procedure, the dataset was partitioned into five subsets, with a distribution ratio of 80\% for training data and 20\% for testing data within each fold.'

Authors' action 2: In our case, our focus is the comparison of the similarity between the signals, creating a model that outputs an rPPG signal similar to the reference PPG in shape. Variations in time and trends are also relevant, but not the focal point of our study. Nevertheless, DDTW or other metrics could be suitable to assess the similarities of the signals from different points of view, and it could be a valid and interesting future research.

Author's action 3: Regarding beat-to-beat HR assessment, it is a very interesting physiological parameter to evaluate, together with others such as oxygen saturation. We have included this information at the end of the discussion as future research: 'As future work, we believe the next steps to follow are the assessment of other physiological parameters, such as beat-to-beat HR assessment and oxygen saturation. This could be useful to identify which physiological parameters can be reliably estimated by the model. Another focal point is to estimate the robustness of the signal produced by the model in different conditions, like the lighting or the recording device used.'

Reviewer #2 (Remarks to the Author):

In the manuscript, authors present a novel machine learning-based approach to improve the construction of remote photoplethysmogram signals from video cameras. The results are interesting and promising, and the paper is well written, but needs some improvements to provide full impact for the readership.

Comments:

- The last paragraph in the introduction should just introduce the study and its aim, and thus most of the information should be moved to the methods and discussion section. Besides, it is well motivated why authors have developed a new method and they clearly explain their motivation to use other metrics than HR in the evaluation.
- Methods, section "rPPG extraction from video": obviously, there is part of a sentence missing. It says "... using the pyVHR framework [27], which includes [28]". If authors want to highlight that something from reference [28] is included, it should be explicitly stated.
- Methods, section "rPPG extraction from video": Please explain what the landmark numbers are meaning. It is not clear at all what authors are referring to.
- Methods, section "rPPG extraction from video": What's reference "2421" in the last sentence? I guess it should read as "21, 24".
- Methods, section "Frequency domain": Why do you not call it $|\Delta HR|$, so readers directly know that it is referring to and can link it to the introduction and discussion, where you state that others compare just HR.
- Methods, section "rPPG extraction from video": Not clear how many datasets and which samples are coming from the subjects and how you end up with the final number. Perhaps you can think of a small figure to illustrate it.
- Methods, section "Statistical Tests": Would not add a description of each of the tests, e.g., Friedman test, but just state in one sentence why and what for you have used the test. Just add a reference to the not so common tests.
- Methods, section "Statistical Tests": Have you applied any correction for multiple testing for the post-hoc tests (e.g., Bonferroni correction)? Would be important! Please state in the text

and would suggest performing them.

- Results: There are many things which should go to the methods section, for example, the description why you have used the specific metrics or the description of the different test cases. In the results section just results should be presented.
- Results, section “Evaluation across datasets”: What do you mean by the sentence “However, the difference in performance is not significant in MR-NIRP and LGI-PPGI on average.”? Please clarify and elaborate.
- Results , Figure 3: Please add N for each dataset since it might influence significances! Furthermore, here correction for multiple testing is important.
- Results, Table 3: I honestly think that lots of information from Table 3 is redundant with information presented in Figure 3 and furthermore I would combine the information with Table 3, where you could easily add a column for the p-values to show their differences to your model – would make reading the results easier. The same holds true for Table 4 and 5.
- Results, Figure 4: Why do you present results here differently than in Figure 3? Would go for the same presentation style as in Figure 3.
- Results, Figure 5: A legend is missing for the figure, thus it is not clear which is the cPPG and which the rPPG.
- Discussion: You mention that there is a disagreement between DTW and RMSE, since RMSE is dependent on possible shifts. A solution could be to align the signal by e.g. using cross-correlation and then calculating the RMSE. Please comment.
- Discussion: Please add thoughts why you think that your model is better than others.
- Discussion: In general, I think it would be important to compare your results with literature. This aspect is very limited in the discussion. E.g., for HR there should be more literature available.
- Conclusion: I would weaken some of your statements, since it is one study where you show that your model outperforms the other models. Thus, saying “the results were optimal” is in my opinion very risky and one should be careful with such strong statements derived on data from one study. I would formulate the statement that results are promising and warrant more research.
- Conclusion: Can you give examples for the future applications?
- English revision is needed for some parts.

Authors’ response: We thank the reviewer for the positive feedback and for accepting the paper for publication.

Authors’ action: The last paragraph of the introduction was shortened as follows:

‘In this study, we present a machine learning–based approach to improve the construction of rPPG signals from video cameras. Our model takes rPPG signals from other models such as chrominance (CHROM)-based rPPG \cite{de2013robust}, local group invariance (LGI) \cite{pilz2018local}, independent component analysis (ICA) \cite{lee1998independent} or plane-orthogonal-to-skin (POS) \cite{wang2016algorithmic} as input and enhances them by comparing them with reference PPG signals. This approach provides a novel method for improving rPPG signal detection using machine learning techniques.’

Authors’ action : Methods: Adapted as suggested: ‘We extracted information from the forehead and cheeks using the pyVHR framework \cite{boccignone2022pyvhr}, which includes

the software MediaPipe for the extraction of RoI from a human face \cite{lugaresi2019mediapipe}.'

Authors' action : Methods: Modified as suggested, and more explicit information about the landmarks:

'Several steps were necessary to extract the rPPG signal from a single video. First, the regions of interest (RoI) were extracted from the face. We extracted information from the forehead and cheeks using the pyVHR framework \cite{boccignone2022pyvhr}, which includes the software MediaPipe for the extraction of RoI from a human face \cite{lugaresi2019mediapipe}. The RoI extracted from every individual were composed of a total of 30 landmarks. Each landmark is a specific region of the face, represented by a number that indicates the location of that region. The landmarks 107, 66, 69, 109, 10, 338, 299, 296, 336, and 9 were extracted from the forehead, the landmarks 118, 119, 100, 126, 209, 49, 129, 203, 205, and 50 were extracted from the left cheek, and the landmarks 347, 348, 329, 355, 429, 279, 358, 423, 425, and 280 were extracted from the right cheek. Every landmark was composed of 30x30 pixels, and the average across the red, green, and blue (RGB) channels was computed for every landmark. The numbers of the landmarks of each area represent approximately evenly spaced regions of that area.'

Authors' action : Methods: References modified as required.

Authors' action : Methods: section "Frequency domain": Metric modified as suggested

Authors' action : Methods: We have introduced a more clear description of the activities and dataset used for training it in the section 'Proposed rPPG construction' of the manuscript':

'The model was trained using data sourced from the PURE dataset. The input data contains information from 10 participants. Each participant was captured across 6 distinct videos, engaging in activities categorized as Steady, Talk, Slow Translation, Fast Translation, Small Rotation, and Medium Rotation. This accounts for a total of 60 videos, with an approximate average duration of 1 minute. Each video was transformed to RGB signals. Then, every RGB set of signals representing a video was subdivided into 10-second fragments, with each fragment serving as a unit for training data. The dataset used to train the model contains a total of 339 such samples.

Because the duration of each video is 10 seconds and the frame rate is 30, each sample is represented by three RGB signals composed of 300 time-steps. The RGB signals, serving as training inputs, underwent a transformation process resulting in the derivation of four distinct signals through the application of the POS, CHROM, LGI, and ICA methods. Consequently, each 10-second segment yielded four transformed signals, which were intended for subsequent utilization as input for the model. Before being fed to the model, data preprocessing was applied to the signals. Then, a 5-fold cross-validation (CV) procedure was conducted. During this procedure, the dataset was partitioned into five subsets, with a distribution ratio of 80% for training data and 20% for testing data within each fold.'

Authors' action : Methods section "Statistical Tests": Description of statistical tests deleted, added Bonferroni correction:

'The Friedman Test \cite{friedman1937use} is appropriate for this study because it evaluates the means of three or more groups. Every group is represented by a model. If the p -value is significant, the means of the groups are not equal. The Nemenyi Test \cite{nemenyi1963distribution} was used to calculate the difference in the average ranking values and then to compare the difference with a critical distance (CD). The general procedure is to apply the Friedmann test to each group and if the p -value is significant, the means of

the groups are not equal. In that case, the Nemenyi test is performed to compare the methods pairwise. The Nemenyi test helps to identify which methods are similar or different in terms of their average ranks. The Bonferroni correction was applied for multiple-comparison correction.'

Authors' action : Results: deleted part that should be in methods: 'To evaluate the performance of the proposed model, we conducted several experiments. The metrics DTW, $\$r\$$, RMSE and $|\Delta HR|$ were implemented, and the evaluation was done across datasets and activities for every dataset.'

Authors' action : Results, section "Evaluation across datasets": it has been rephrased more clearly, so that the reader can understand the context and figure 3. 'However, even though our model shows the best $\$r\$$ for the datasets MR-NIRP and LGI-PPGI, the difference is not significant when compared to other models such as CHROM or POS.'

Authors' action : Results, figure 3: included in caption 'Taking into account the Bonferroni correction, the adjusted significance level is 0.003. The total number of samples is 339 for PURE, 251 for LGI-PPGI and 187 for MR-NIRP.'

It has been also added to figure 4: 'Boxplot of the $\$r\$$ coefficient and DTW for the different methods. The analysis was done across activities. The $\$p\$$ -value of every method against our model is given above the boxes. The $\$p\$$ -values were obtained by applying the Friedman and post hoc Nemenyi tests. Taking into account the Bonferroni correction, the adjusted significance level is 0.003. The total number of samples is 339 for PURE, 251 for LGI-PPGI and 187 for MR-NIRP.'

Authors' action : Results, tables: to avoid showing redundant information, the tables have been moved to the appendix.

Authors' action: Results, figure 4: The style of the figure has been changed

Authors' action: Results, figure 5: The legend has been added

Authors' action: Discussion: The comment has been added: 'One potential resolution could involve signal alignment through techniques like cross-correlation, followed by RMSE calculation. This alignment procedure could alleviate the current inconsistency and provide a more unified perspective on signal evaluation. Further exploration and validation of this alignment strategy could shed light on its effectiveness in resolving the observed disagreement between DTW and RMSE.'

Authors' action: Results: added information on why our model is the best alternative: 'Through the gathered information and results shown, our model outperforms the other methods in most of the settings. When it comes to the performance across activities, our model had the best performance for the six activities and both metrics DTW and $\$r\$$, with the exemption of the activity Gym for the metric $\$r\$$, where POS is the best model. In the analysis of performance across datasets, our model shows the lowest DTW for every dataset. It also outperforms other methods for the metrics $\$r\$$ and RMSE, except for the MR-NIRP dataset, where the advantage is not always significant.'

Authors' action: Discussion: we modified the comparison of the HR results with other studies, having a total of four results from other studies to compare with:

'While most previous studies \cite{hu2021eta,song2021pulsegan} have focused on HR, as discussed above, when a model is created only to detect HR, information about other frequencies and features of the wave is lost, such as the diastolic peak. For the PURE dataset, the results of ETA-rPPG \cite{hu2021eta} in HR estimation are 0.34 $|\Delta HR|$, which is a better result than ours (0.52). For Siamese-rPPG \cite{tsou2020siamese}, the $|\Delta HR|$ is 0.51, for DeepPhys \cite{chen2018deepphys} 0.83, and for HR-CNN \cite{vspetlik2018visual} 1.84.'

Authors' action: Conclusion: statement modified: 'The results of our proposed model are promising in comparison with the best-performing traditional methods, including POS, LGI, and CHROM and warrant more research.'

Authors' action: Conclusion: modified as suggested: ' This offers a non-invasive alternative with many future applications, such as Health Monitoring and Telemedicine, and Driver Monitoring and Biometric Authentication'.

Reviewer #3 (Remarks to the Author):

Dear Dr. Ontiveros, Dr. Elgendi and Dr. Menon,

The manuscript 'A Machine Learning-Based Approach for Constructing Remote Photoplethysmogram Signals from Video Cameras' proposes a novel machine learning-based approach to improve the construction of rPPG signals from video cameras. The model utilizes CHROM, LGI, ICA, POS, and GRGB rPPG methods. Three public datasets with activities were used, of which one dataset was used to train and test the model. Comparison was done using Pearson correlation, DTW and BPM difference. The results demonstrated that the proposed method can construct an rPPG signal resembling cPPG signal recorded from the fingertip, offering a more non-invasive alternative for PPG recordings. The performance of the proposed model was also better compared with the best-performing previous methods.

The methods are reported clearly to allow replication and the analysis seems to be sound. The manuscript has novelty value in the research field of PPG analytics and machine learning. The current proposed method takes the morphology of the PPG signal into account better in contrast to previous most studies where the metrics are calculated from the predicted HR or ground truth HR. The conclusions are supported by the data. The results are interesting, and I believe that rPPG technology could be utilized in the future perhaps for vital sign contact-free monitoring e.g. HR, rhythm, respiratory rate.

I think the manuscript could be suitable for publication and would interest the readers of Communications Medicine. However, I have some detailed questions or comments below. Perhaps minor improvements could be considered, but the manuscript is very good in its current state. I sincerely hope that my comments can help the authors to improve the manuscript, and the editors to make a decision on the manuscript.

With best regards,
Dr. Väliäho

Q1. Page 12, Conclusion. “We have successfully developed a method that can construct an rPPG signal that resembles the PPG signal from the fingertip. This offers a non-invasive alternative with many future applications.” In medical sciences, ‘invasive’ techniques usually involve introduction of instruments into the body, and ‘non-invasive’ do not. E.g. ultrasound technique with the probe or contact PPG sensor placed against the skin of the patient are considered as non-invasive methods, and e.g. angiography with instruments inserted through the skin into the arteries are considered as invasive method. I would say that the proposed rPPG is non-invasive alternative indeed, but so is contact PPG. The current phrasing is a bit confusing possibly suggesting that fingertip PPG would be something else than non-invasive. Could this be made a bit clearer? Or perhaps the expression contact-free could be considered.

Q2. Table 4, Table 5, and Figure 5. These are now presented in the Discussion section. I recommend repositioning these in the Results section.

Q3. Page 11, “In most studies, these metrics are calculated..” Please provide references at least to some of these studies. On page 12 “While most previous studies..” references are provided.

Q4. Some publications could be worth looking into to discuss the results of the current manuscript in comparison. It has been demonstrated that atrial fibrillation arrhythmia can be detected from contact-free facial rPPG obtained from video signals (Martinez-Delgado et al., 2022, Yan et al., 2020), and with pretrained deep convolutional network (Yan et al., 2020). In addition, Charlton et al. published 2022 an interesting editorial for establishing best practices in PPG signal acquisition and processing.

Q5. On page 1, Introduction. Artifacts of rPPG signals are mentioned here. Were there issues with the current rPPG data? And how do the artifacts in rPPG compare to cPPG? This is not discussed in the Discussion section, I suggest adding some discussion about this as it is mentioned in the introduction as a problem of rPPG.

Q6. Although the manuscript focuses on machine learning, could the authors please consider adding some examples of potential applications of rPPG in the Discussion? E.g. in clinical medicine or physiology. I think rPPG could be utilized in the future to monitor in-hospital patients, e.g. respiratory rate, rhythm, heart rate.

References:

Charlton PH, Pilt K, Kyriacou PA. Establishing best practices in photoplethysmography signal acquisition and processing. *Physiol Meas*. 2022 May 25;43(5):050301. doi: 10.1088/1361-6579/ac6cc4. PMID: 35508148; PMCID: PMC9136485.

Martinez-Delgado GH, Correa-Balan AJ, May-Chan JA, Parra-Elizondo CE, Guzman-Rangel LA, Martinez-Torteya A. Measuring Heart Rate Variability Using Facial Video. *Sensors (Basel)*. 2022 Jun 21;22(13):4690. doi: 10.3390/s22134690. PMID: 35808182; PMCID: PMC9269597.

Yan BP, Lai WHS, Chan CKY, Au ACK, Freedman B, Poh YC, Poh MZ. High-Throughput, Contact-Free Detection of Atrial Fibrillation From Video With Deep Learning. *JAMA Cardiol*. 2020 Jan 1;5(1):105-107. doi: 10.1001/jamacardio.2019.4004. PMID: 31774461; PMCID: PMC6902123.

Authors' response: We thank the reviewer for the positive feedback and for accepting the paper for publication.

Authors' action:

Q1: The expression contact-free was considered.

Q2: Table 4 and 5 have been moved to the appendix and the figure has been repositioned as well.

Q3: references added: 'In most studies \cite{hu2021eta, tsou2020siamese, lu2021dual}'

Q4: The references have been added in the introduction: '...HR and HRV extracted from cPPG \cite{niu2020video, gudi2020real, haugg2023grgb, martinez2022measuring}. Other authors have focused on improving BP \cite{schrumpf2021assessment}, oxygen saturation measurements \cite{kim2021non} or fibrillation arrhythmia \cite{yan2020high}.'

Q5: Added to introduction: 'For instance, in the dataset LGI-PPGI, the videos in which the subjects talk are recorded outdoors, impacting the quality of the rPPG. When comparing factors in rPPG to conventional cPPG, certain distinctions emerge. While both methods are susceptible to motion-related artifacts, rPPG faces additional challenges due to the reliance on non-contact measurements, making it more sensitive to environmental changes and subject movement.'

Q6: As reviewer 2 also suggested, we included some applications (in the conclusion): 'This offers a non-invasive alternative with many future applications, such as Health Monitoring and Telemedicine, and Driver Monitoring and Biometric Authentication'